# *Lactobacillus plantarum* Zhang-LL Inhibits Colitis-Related Tumorigenesis by Regulating Arachidonic Acid Metabolism and CD22-Mediated B-Cell Receptor Regulation

**DOI:** 10.3390/nu15214512

**Published:** 2023-10-25

**Authors:** Jingxin Zhu, Wenbo Liu, Zheng Bian, Yumeng Ma, Zixin Kang, Junhua Jin, Xiangyang Li, Shaoyang Ge, Yanling Hao, Hongxing Zhang, Yuanhong Xie

**Affiliations:** 1Beijing Laboratory of Food Quality and Safety, Beijing Key Laboratory of Agricultural Product Detection and Control of Spoilage Organisms and Pesticide Residue, College of Food Science and Engineering, Beijing University of Agriculture, Beijing 102206, China; xin202130611006@163.com (J.Z.); l18811301300@163.com (W.L.); bianzheng0311@163.com (Z.B.); m15210115481@163.com (Y.M.); 13126986155@163.com (Z.K.); jinjunhua002008@163.com (J.J.); lxy2002cn@163.com (X.L.); 2Beijing HEYIYUAN BIOTECHNOLOGY Co., Ltd., Beijing 100088, China; geshaoyang@foxmail.com; 3Key Laboratory of Functional Dairy, Department of Nutrition and Health, Co-Constructed by Ministry of Education and Beijing Government, China Agricultural University, Beijing 100190, China; haoyl@cau.edu.cn

**Keywords:** colorectal cancer, *Lactobacillus plantarum* Zhang-LL, gut microbiota, arachidonic acid metabolism, CD22

## Abstract

Colorectal cancer (CRC) is a significant health concern and is the third most commonly diagnosed and second deadliest cancer worldwide. CRC has been steadily increasing in developing countries owing to factors such as aging and epidemics. Despite extensive research, the exact pathogenesis of CRC remains unclear, and its causes are complex and variable. Numerous in vitro, animal, and clinical trials have demonstrated the efficacy of probiotics such as *Lactobacillus plantarum* in reversing the adverse outcomes of CRC. These findings suggest that probiotics play vital roles in the prevention, adjuvant treatment, and prognosis of CRC. In this study, we constructed a mouse model of CRC using an intraperitoneal injection of azomethane combined with dextran sodium sulfate, while administering 5-fluorouracil as well as high- and low-doses of *L. plantarum* Zhang-LL live or heat-killed strains. Weight changes and disease activity indices were recorded during feeding, and the number of polyps and colon length were measured after euthanasia. HE staining was used to observe the histopathological changes in the colons of mice, and ELISA was used to detect the expression levels of IL-1β, TNF-α, and IFN-γ in serum. To investigate the specific mechanisms involved in alleviating CRC progression, gut microbial alterations were investigated using 16S rRNA amplicon sequencing and non-targeted metabolomics, and changes in genes related to CRC were assessed using eukaryotic transcriptomics. The results showed that both viable and heat-killed strains of *L. plantarum* Zhang-LL in high doses significantly inhibited tumorigenesis, colon shortening, adverse inflammatory reactions, intestinal tissue damage, and pro-inflammatory factor expression upregulation. Specifically, in the gut microbiota, the abundance of the dominant flora *Acutalibacter muris* and *Lactobacillus johnsonii* was regulated, PGE2 expression was significantly reduced, the arachidonic acid metabolism pathway was inhibited, and CD22-mediated B-cell receptor regulation-related gene expression was upregulated. This study showed that *L. plantarum* Zhang-LL live or heat-inactivated strains alleviated CRC progression by reducing the abundance of potentially pathogenic bacteria, increasing the abundance of beneficial commensal bacteria, mediating the arachidonic acid metabolism pathway, and improving host immunogenicity.

## 1. Introduction

Colorectal cancer (CRC) is the second leading cause of cancer-related deaths. Between 1990 and 2019, the global incidence of CRC increased from 840,000 to 2.17 million, and the number of deaths increased from 510,000 to 1.09 million [1]. Alarmingly, the disease burden is shifting towards younger individuals, with one out of every five new cases occurring in people under the age of 50 [2,3]. Presently, the exact pathogenesis of CRC is unknown, and ultra-processed foods [4], alcohol [5], processed red meat intake [6], smoking [7], obesity [8], lifestyle [9], and family genetic history [10] are considered to be highly correlated with CRC incidence. Compared to traditional chemotherapy drugs [11], new treatments such as gene therapies, immunotherapies, and targeted gene therapies are in the clinical trial stage, and many adverse side effects are difficult to avoid [12].

CRC is a malignant tumor with the highest incidence in the digestive system. As such, its relationship with the gut microbiota has received widespread attention [13]. In patients who underwent Whipple surgery, the tumor microbiome represented approximately 25% of the gut microbiome, and the tumor microbiome was absent from normal adjacent tissues [14]. In addition, microorganisms within tumors regulate tumor initiation and progression and alter the tumor response to chemotherapy and immunotherapy [15]. Based on these findings, different active substances or fecal microbiota transplantation (FMT) have been used to intervene in the manipulation of CRC progression using the gut microbiota and metabolites [16,17,18].

Probiotics are live microorganisms that are beneficial to the host at adequate doses [19]. Postbiotics are defined as preparations of inanimate microorganisms and/or their components that provide health benefits to the host [20]. Postbiotics include inactivated microbial biomass, whole microbial cells, cell components and fragments, and/or mixtures of post-fermentation metabolites. 

There is substantial evidence that specific probiotics or postbiotics and beneficial intestinal commensal bacteria affect cancer cell proliferation, apoptosis, the regulation of gut microbiota, the maintenance of intestinal barrier function, the direct regulation of cancer-related signaling pathways, and the stimulation of host immune responses [21,22,23]. Bioactive substances include exopolysaccharides, peptides, and short-chain fatty acids [17,24,25], but their specific mechanisms have not been elucidated, and the results are highly heterogeneous. Probiotics such as *Lactobacillus* and *Bifidobacterium* are frequently chosen for testing based on previous clinical trial results to investigate their mechanism of action in alleviating CRC [26], enrich their application perspectives in prevention and adjuvant therapy, improve prognosis, and provide a theoretical basis and potential practical direction for the future development of complex probiotic nutritional supplements.

However, most in vitro studies are unable to recreate the actual situation in vivo, and clinical trials can only select probiotic strains with anti-tumor activity, which limits the ability to clarify the mechanism of action. *L. plantarum* Zhang-LL was isolated from traditional Chinese fermented meat products and produced class IIa bacteriocins, which have antibacterial activity against a variety of Gram-positive and Gram-negative bacteria [27]. Previously, we found that *L. plantarum* Zhang-LL and its inactivated strains alleviated symptoms in rats with chronic ulcerative colitis (UC) by modulating gut microbiota, maintaining the intestinal barrier, and downregulating the expression of pro-inflammatory factors [28]. Furthermore, bacteriocin BM-1 was found to reduce the viability of SW480 human CRC cells by inducing caspase-dependent apoptosis [29]. 

In this study, the anti-CRC effect of the probiotic *L. plantarum* Zhang-LL and heat-inactivated strains was explored using the azomethane/dextran sodium sulfate method to induce CRC in mice, with fluorouracil as a positive control. By analyzing changes in the gut microbiota, derived metabolites, and host transcription levels under different treatments, the regulatory relationship between specific gut microbiota structures and host gene expression was revealed.

## 2. Materials and Methods 

### 2.1. Strain Preparation

*L. plantarum* Zhang-LL (CGMCC 6939) frozen at −80 °C was activated at 37 °C in de Man, Rogosa, and Sharpe (MRS) broth (LuQiao Company, Beijing, China) for 12 h, and the strain concentration was adjusted to 1 × 10^9^ CFU/mL after three passages. The strain suspension was centrifuged at 8000× *g* at 4 °C for 15 min. After discarding the supernatant, one-tenth of the original volume of 0.85% normal saline was added, and a 1 × 10^10^ CFU/mL strain suspension was obtained after resuspending. Heat-killed strain suspensions were prepared by heat treatment (80 °C, 20 min) for subsequent experiments.

### 2.2. Animal and Experimental Design

Fifty-six SPF-rated 6-week-old C57BL/6N healthy male mice were obtained (Charles River, Beijing, China) and housed in the SPF-level experimental animal room at the Beijing University of Agriculture at 22 °C ± 2 °C with 50% ± 10% humidity and a 12 h/12 h light/dark cycle, in accordance with the charter of the Animal Ethics Committee of the Beijing University of Agriculture (BUA2023069). The mice were randomly divided into seven groups of eight according to their body weight. During adaptive feeding in week one of the experiment, the mice were allowed to eat and drink water freely. The experiment began in week two, and all mice with the exception of those in the control group (group C) were injected intraperitoneally with 10 mg/kg azomethane (AOM; Sigma-Aldrich, St. Louis, MO, USA) on day one. A week later, the normal drinking water of mice was treated with 2% dextran sodium sulfate (DSS, MP Biomedicals, CA, USA) for five days, and then mice were allowed to recover for 14 days (group M). The water-change operation was repeated twice followed by a two-week recovery period each time. At the same time, different treatments were carried out. Mice in the positive control group (group F) were intraperitoneally injected with 20 mg/kg/3d of the chemotherapy drug fluorouracil (5-FU, Sigma-Aldrich, St. Louis, MO, USA). Mice in the live bacteria group (group L_L and L_H, respectively) were gavaged with 200 μL of a 1 × 10^9^ or 1 × 10^10^ CFU/mL live strain suspension daily, and mice in the heat-inactivated bacteria group (group HK_L and HK_H, respectively) were gavaged with 200 μL of a 1 × 10^9^ or 1 × 10^10^ CFU/mL heat-killed strain suspension daily. The body weight of all mice was recorded weekly during rearing. After the 78th day, cervical dislocation was performed, and stool and intestinal content samples were collected and stored at −80 °C. The colonic tissue was washed with phosphate-buffered saline (PBS), the colon length and the number of tumors were recorded, and the colon was cut into two lengthwise segments; one segment was stored at −80 °C and the other was fixed in a 4% phosphate-buffered formaldehyde solution.

### 2.3. Disease Activity Index (DAI) Evaluation

The DAI score is used as an indicator to evaluate the degree of inflammation and is determined by the rate of weight loss, fecal viscosity, and fecal occult blood status. The fecal occult blood status was scored using the Fecal Occult Blood Qualitative Test Kit (Leagene Biotechnology, Beijing, China) as follows. Weight loss was scored as: 0: <1%; 1: ≥1% and <5%; 2: ≥5% and <10%; 3: ≥10% and <15%; and 4: ≥15%. Fecal viscosity was scored as: 0: normal; 2: loose stool; and 4: diarrhea. Fecal occult blood was scored as: 0: No color development within two min; 1: After 10 s, there is a change from colorless to light purple or purple; 2: Color is light purple at first and gradually becomes distinctly purple; 3: Immediately appears purple at first and gradually becomes dark purple; and 4: Immediately appears dark purple.

### 2.4. Histopathology

The colon tissue was fixed at 4 °C for 24 h. After washing with PBS (10 mM, pH 7.4), sections were dehydrated using an ethanol gradient (50, 70, 95, and 100% ethanol) and embedded in paraffin. The paraffin block was then cut into 5 μm sections, stained with hematoxylin and eosin (HE), and finally analyzed for histological changes in colon tissue.

### 2.5. Gut Microbiota Analysis

Six biological replicates were selected for mice intestinal contents samples. The raw sequencing reads were deposited in the NCBI Sequence Read Archive (SRA) database (Accession Number: SRP443480). Bioinformatic analysis of the gut microbiota was performed using the Majorbio Cloud platform (https://cloud.majorbio.com, accessed on 7 September 2023). The specific experimental method refers to the study of Han et al. [30].

### 2.6. Untargeted Metabolomics

Six biological replicates were selected from the fecal samples of the mice, and the metabolites were extracted after grinding occurred in liquid nitrogen. This experimental method refers to the study of Du et al. [31].

### 2.7. Eukaryotic Transcriptomics

Four biological replicates were selected from mice intestinal tissue samples. The raw sequencing reads were deposited in the NCBI Sequence Read Archive (SRA) database (Accession Number: SRP441958). The experimental method of this part of the study by Liu et al. [32] is referenced.

### 2.8. Cytokine Assay

Serum and intestinal cytokine levels were measured using ELISA. Prostaglandin E2 (PGE2), Interleukin-1β (IL-1β), Tumor Necrosis Factor-α (TNF-α), and Interferon-γ (IFN-γ) assay kits (Nanjing Jiancheng, Nanjing, China) were used as directed by the manufacturer.

### 2.9. Data Analysis

All data are presented as mean ± standard deviation, with significant differences of *p* < 0.05. One-way ANOVA was performed using IBM SPSS Statistics 23 and Origin 2021, and post hoc analysis was performed using the LSD, Tukey, and Tamhane T2 tests.

## 3. Results

### 3.1. High Doses of Zhang-LL Bacteria Mitigate CRC Progression in AOM/DSS Mice

Based on the AOM/DSS mouse model, treatment groups of high and low doses of Zhang-LL live bacteria (L_H and L_L, respectively) and the heat-killed strain (HK_H and HK_L, respectively) were designed and compared with the active drug treatment group (group F) (Figure 1A) to determine the dose-response relationship of Zhang-LL and the extent of anti-tumor activity. During the modeling process, mice that drank DSS (group M) experienced a significant weight loss (*p* < 0.001) compared to the control group (group C) during the dosing cycle, and their weight recovered during the recovery period. L_H, HK_H, L_L, and HK_L reversed the large changes in body weight, with significant effects observed in the high-dose groups (*p* < 0.05) compared to those in group C (Figure 1B). The DAI was assessed weekly to monitor symptoms in the mice, including the rate of weight loss, fecal viscosity, and fecal occult blood. Compared to group C, group M had a significantly increased DAI (*p* < 0.01); L_H and HK_H partially alleviated DAI in AOM/DSS mice, and L_L and HK_L had little effect on DAI changes (Figure 1C).

Changes in polyp number and colon length are important features of colorectal progression. Compared with group M, groups L_H, HK_H, and F significantly reduced the number of polyps in mice (*p* < 0.001), whereas the effects of L_L and HK_L in the low-dose group were not obvious (Figure 1D). Compared to group C, mice in group M showed significant colon shortening (*p* < 0.01); L_H and HK_H could prevent colonic changes, but only group L_H had a significant protective effect (*p* < 0.05) and approached the therapeutic level of group F (*p* < 0.01) (Figure 1E,F). HE staining showed that large tumor cell masses were visible in the mucosal layer of the intestinal tissue in group M and tumor cells were arranged in glandular tubes, whereas groups L_H and HK_H only had large tumor cell masses or multiple ulcers locally, and no tumor cell masses were found in individual samples. No tumor cell masses were found in group F, indicating a positive therapeutic effect, and group C had a normal morphological structure and no obvious inflammatory changes (Figure 1G). HE staining also showed that compared to group C, the number of goblet cells decreased significantly in group M (*p* < 0.001), and only HK_H prevented goblet cell loss (*p* < 0.05); L_H and F had little effect (Figure 1H). The expression levels of inflammatory factors IL-1β, TNF-α, and IFN-γ in the serum of mice were detected, and the serum contents of IL-1β and TNF-α in group M were significantly higher than those in group C (*p* < 0.001); L_H significantly reduced the expression of two inflammatory factors (*p* < 0.01), and HK_H only reduced IL-1β expression (*p* < 0.05). IFN-γ expression in the serum of only group F was significantly lower than that of group M (*p* < 0.01), and F significantly reduced the expression of IL-1β and TNF-β (*p* < 0.05) (Figure 1I–K).

These results indicate that L_H and HK_H alleviated CRC progression in AOM/DSS mice, whereas L_L and HK_L had no significant effect; therefore, the high-dose groups were selected for additional experiments.

### 3.2. L_H and HK_H Regulate Gut Microbiota Structure in AOM/DSS Mice

We speculated that Zhang-LL gradually regulated the structure of mouse gut microbiota at different time points to exert anti-tumor effects. We compared the changes in gut microbiota diversity in different treatment groups at weeks 0 (W_0_), 7 (W_7_), and 13 (W_13_) by 16S rDNA sequencing combined with biological diversity analysis. A rarefaction curve analysis of the OTUs showed that the gut microbiota were sequenced deep enough to cover the microbial diversity of the sample (Figure 2A). The first collection of mouse feces at W_0_ was not subjected to any treatment, and the phyla between different groups showed a similar composition, with *Firmicutes* and *Bacteroidetes* as the main flora (Figure 2(B1)); there was no significant difference in α diversity. W_7_ was the time point in the mold-making process (Figure 2(B2)), and it was found that the ratio of *Firmicutes* to *Bacteroidetes* increased in group M, but the difference was not significant. At W_13_, after the euthanasia of mice (Figure 2(B3)), significant changes in *Firmicutes* and *Bacteroidetes* were observed (Figure 2C,D), indicating gut microbiota dysbiosis in group M during the modeling process, while L_H and HK_H partially reversed intestinal dysbacteriosis. Compared with the other treatment groups, the Sobs, Chao, ACE, Simpson, and Shannon indices of group F decreased, but only the Sobs index decreased significantly (*p* < 0.05) (Figure 2E), indicating that the administration of chemotherapy drugs reduced the abundance of gut microbiota.

In the PLS-DA analysis, there was a transition trend, as L_H- and HK_H-processed samples at W_0_, W_7_, and W_13_ clustered at different locations (Figure 2F). At W_13_, groups M and C appeared at different positions, representing the gut microbiota of AOM/DSS mice with different structures. Groups L_H and HK_H reshaped the gut microbiota, and the two groups were relatively close to each other, indicating an attempt to restore the microbiota structure to its original state, whereas the intervention in group F demonstrated a completely different microbiota structure (Figure 2G,H).

At the genus level, the differences between groups C, F, L_H, HK_H, and M were compared separately using two groups for comparison. Potential pathogenic genera in group C included *Desulfovibrio* and *Romboutsia* in significantly lower abundance compared to that of group M (*p* < 0.05), and beneficial bacteria, such as *Alloprevotella*, were significantly more abundant than in group M (*p* < 0.05) (Figure 2I). Similarly, the abundances of the beneficial bacteria *Bifidobacterium* and *Weissella* in groups F, L_H, and HK_H were significantly higher than those in group M (*p* < 0.05) (Figure 2J–L). Next, we focused on the enrichment of bacterial species after F, L_H, and HK_H interventions and identified the dominant flora. *Bifidobacterium animalis* and *Bifidobacterium pseudolongum* were significantly increased in abundance in group F (*p* < 0.05) (Figure 2M). Group L_H showed a significant increase in the abundance *of L. plantarum* and *Lactobacillus paracasei* at the species level (*p* < 0.05), whereas the abundances of *Enterococcus faecium* and *Bacteroides nordii* decreased significantly (*p* < 0.05) (Figure 2N). In addition, different outcomes were observed after HK_H intervention, with the abundance of *Acutalibacter muris* and mouse gut metagenome *Enterorhabdus* increasing significantly at the species level compared to group M (*p* < 0.05), while the abundance of *Lactobacillus murinus* and *Lactobacillus johnsonii* decreased significantly (*p* < 0.05) (Figure 2O).

In conclusion, the gut microbiota plays an important role in CRC progression in AOM/DSS-treated mice. L_H and HK_H reshaped the structure of the gut microbiota, reduced the abundance of potentially pathogenic bacteria, and increased the abundance of beneficial commensal flora in some species of the genera *Lactobacillus* and *Acutalibacter*.

### 3.3. L_H and HK_H Alter Gut Microbiota-Derived Metabolites in AOM/DSS Mice

Experimental data have shown that *Lactobacillus* plays an important role in reshaping the gut microbiota. To further elucidate the effects of L_H and HK_H on the gut microbiota-derived metabolites of AOM/DSS mice, the intestinal contents of the mice were collected for LC-MS/MS analysis, and specific metabolic process changes were explored using bioinformatics analysis. Most metabolites detected were amino acids, phospholipids, and polysaccharides. KEGG pathway enrichment analysis found that 100 metabolites were enriched in lipid metabolism and 101 metabolites were enriched in amino acid metabolism; therefore, we focused on metabolites related to these processes. Compared to group M, 43 metabolites were upregulated and 81 were downregulated in group F (Figure 3A). In group L_H, 69 metabolites were upregulated and 53 were downregulated (Figure 3B). In group HK_H, 110 metabolites were upregulated and 115 were downregulated (*p* < 0.05; VIP > 1; up-down-regulation by a multiplier of 1) (Figure 3C). However, short-chain fatty acids associated with CRC progression were not found in the differential metabolites enriched in different metabolic pathways.

Furthermore, KEGG pathway enrichment analysis showed that only 5 differential metabolites in group F were enriched in nucleotide metabolism (Figure 3D), whereas 13 and 16 differential metabolites were co-enriched in the lipid metabolism pathway in groups L_H and HK_H, respectively (Figure 3E,F). Importantly, the KEGG enrichment heat map showed that all differential metabolites were co-enriched in lipid metabolism processes in L_H and HK_H, and the common metabolic pathway associated with lipid metabolism was the arachidonic acid metabolism pathway (map00590); four and five differential metabolites were enriched in groups L_H and HK_H, respectively (Figure 3J). In this pathway, the differential metabolites were predominantly enriched in metabolic processes associated with prostaglandin E2 (PGE2) and its isomers (Figure 3G–I). Changes in the PGE2 levels in the intestinal contents were measured to validate the metabolomic results. The ELISA results showed a significant increase in PGE2 content in group M (*p* < 0.05), which decreased to varying degrees in both groups L_H and HK_H (*p* < 0.01) (Figure 3K). These results indicate that L_H and HK_H regulate the gut microbiota and inhibit arachidonic acid catabolic processes in the tumor microenvironment to mitigate CRC progression in AOM/DSS mice.

### 3.4. Effects of L_H and HK_H on Gene Expression in Colon Tissues of AOM/DSS Mice

PGE2 downregulation was shown to be critical for mitigating CRC progression in AOM/DSS mice, and if combined with host gene expression, it could be used to study the mechanism from the phenotypic to the molecular level. Mouse colon tissues were collected for eukaryotic transcriptomic analysis combined with bioinformatics analysis to identify DEGs. The results of the volcano map showed that a total of 70 genes were upregulated and 82 genes were downregulated in group F, 51 genes were upregulated and 105 genes were downregulated in group L_H, and 415 genes were upregulated and 328 genes were downregulated (differential multiples ≥ 2) in group HK_H (Figure 4A–C). Based on gene function enrichment analysis in the Reactome database, group F was significantly enriched in DNA damage/telomere stress-induced aging pathways by inhibiting essential biosynthetic processes or by incorporating macromolecules (such as DNA and RNA) and hindering the normal function of cancer cells [11]. A total of 32 and 33 immune system-related pathways were found to be enriched in groups L_H and HK_H, respectively (*p* < 0.05). By screening enrichment factors, we found that 10 and 33 genes were co-enriched in the process of CD22-mediated B-cell receptor (BCR) regulation in groups L_H and HK_H, respectively (Figure 4D–G). In addition, when gene function enrichment analysis was used in the KEGG database, there was no enriched pathway in group F, while pathways associated with CRC progression (*p* < 0.05) were enriched in groups L_H and HK_H, including the Toll-like receptor, NOD-like receptor, mTOR, and Hippo signaling pathways (Figure 4H).

In summary, gut microbiota regulation by L_H and HK_H may affect CD22-mediated BCR regulatory processes in the tumor immune microenvironment to alleviate CRC progression in AOM/DSS mice.

### 3.5. Multi-Omics Joint Analysis of the Effects of L_H and HK_H on Gut Microbiota, Metabolites, and Gene Expression in AOM/DSS Mice

This study aimed to use the arachidonic acid metabolic pathway as a bridge to reveal the bidirectional regulatory relationship between specific gut microbiota structures and host gene expression following L_H and HK_H treatment. Pearson correlation analysis was used to establish the correlation between the gut microbiota and arachidonic acid metabolism as well as between the CD22-mediated BCR regulation process and arachidonic acid metabolism.

Based on Pearson’s correlation analysis, *Akkermansia muciniphila* and *Burkholderiales* bacterium YL45 in group L_H showed a strong positive correlation (>0.9) with prostaglandin-c2, 15H-11, 12-EETA, prostaglandin A2, and prostaglandin B2 enriched in the arachidonic acid metabolic pathway. *L. murinus*, *Clostridium* sp. Culture-27, and *A. muris* were strongly negatively correlated with four metabolites (<−0.7) (Figure 5A). Moreover, group HK_H showed that *L. johnsonii*, *Enterorhabdus* mouse gut metagenome, and *Clostridium* sp. Culture-27 were strongly positively correlated (>0.7) with five metabolites, including 20-hydroxyeicosatetraenoic acid, prostaglandin-C2, 15H-11, 12-EETA, prostaglandin A2, and prostaglandin B2 enriched in the arachidonic acid metabolic pathway. *Bacteroides vulgatus* and *A. muris* were strongly negatively correlated (<−0.8) with five metabolites (Figure 5B). Interestingly, *L. johnsonii*, *Enterorhabdus* mouse gut metagenome, and *A. muris* in group HK_H showed strains consistent with the dominant flora. Although there was no specific strain in group L_H that was consistent with the dominant flora, a strong positive correlation with *Akkermansia* (>0.8) and a strong negative correlation with *Lactobacillus* (<−0.8) were found in the correlation analysis between arachidonic acid metabolism and genus-level dominant flora (Figure 5C).

Correlation analysis between the CD22-mediated BCR regulation process and arachidonic acid metabolism showed that gkv1-110 had a strong positive correlation (>0.9) with four arachidonic acid metabolism-related metabolites in group L_H, whereas Ighv1-59 showed a strong negative correlation (<−0.6) (Figure 5D). A strong positive correlation was observed with ghv5-17 (>0.9), with five metabolites enriched in arachidonic acid metabolism, whereas Ighv1-26, Ighv1-82, Igkv8-21, Ighv1-52, Ighv14-2, Ighv1-55, Ighv8-8, and Ighv1-47 showed strong negative correlations (<−0.9) (Figure 5E).

In summary, combined with changes in the abundance of dominant flora and Pearson association analysis, *Lactobacillus* in group L_H and *A. muris* and *L. johnsonii* in group HK_H promoted the decomposition process of arachidonic acid (Figure 5F–H) and upregulated the expression of relevant genes in the process of CD22-mediated BCR regulation to alleviate CRC progression in AOM/DSS mice.

## 4. Discussion

The biological properties of probiotics provide potential prevention strategies for CRC and may serve as an important component of future interventions for the progression of CRC. Severe malnutrition is common in patients undergoing surgery for gastrointestinal cancer and is a risk factor for death up to 30 days after CRC surgery, and it has been shown that nutritional interventions can improve early outcomes following surgery for gastrointestinal cancers [33]. However, chemotherapy drugs have adverse effects, such as decreasing patient survival, destroying intestinal tissue morphology to induce hepatotoxicity, altering intestinal immune status, and disrupting the balance of the gut microbiota [34]. By regulating the gut microbiota, oral probiotics reduce chemotherapy drug resistance and improve the effect of chemotherapy on adjuvant CRC [35,36]. However, the anti-tumor mechanisms of probiotics have not been elucidated and are insufficient to guide future clinical trials. As a result, there is an urgent need for probiotics with anti-CRC effects so that patients can be provided with personalized nutrition plans to meet their unique needs.

Our study showed that *L. plantarum* Zhang-LL effectively alleviated CRC progression in AOM/DSS mice by regulating the gut microbiota and derived metabolites, thereby improving host immunogenicity. Tumorigenesis, colon shortening, adverse inflammatory symptoms, intestinal tissue damage, and intestinal inflammation were all significantly inhibited by Zhang-LL by reducing the abundance of potentially pathogenic bacteria, increasing the abundance of beneficial commensal bacteria, enriching the arachidonic acid metabolism flora, and significantly reducing the expression of the key factor PGE2 in the arachidonic acid metabolism pathway. Simultaneously, the expression of genes related to arachidonic acid metabolism and the CD22-mediated BCR regulation process was significantly upregulated, slowing CRC progression.

Zhang-LL significantly inhibited the AOM/DSS-induced dysbacteriosis of gut microbiota and reshaped the intestinal environment by changing the number of OTUs, α-diversity (Appendix A), β-diversity, and the enrichment of dominant flora at the genus and species levels. Zhang-LL colonized the gut of mice to some extent, but the diversity results indicated that the overall diversity of the microbiota was not significantly changed. In this regard, oral probiotics aim to significantly alter the activity of intestinal commensal bacteria rather than simply reshaping the intestinal microbiota. Specifically, an elevated ratio of *Firmicutes* to *Bacteroidetes* at the phylum level indicates intestinal dysbacteria [25,37], and Zhang-LL is effective in reversing this change. At the genus level, the abundance of beneficial commensal bacteria, including *Alloprevotella*, *Bifidobacterium*, and *Weissella*, increased significantly in the gut. Changes at the species level were equally significant, with specific flora including *B. animalis*, *L. plantarum*, and *A. muris* involved in gut microbiota regulation.

At the genus level, *Desulfovibrio* and *Romboutsia* increased in abundance after AOM/DSS induction and were also observed when sequencing 16S rRNA in fecal samples from high-fat diet-fed rats and patients with CRC liver metastases [38,39], which is consistent with previous studies [40]. The increased abundance of the beneficial bacteria *Enterorhabdus*, *Bifidobacterium*, and *Weissella* and the decreased abundance of the potentially pathogenic bacteria *Enterococcus* in all intervention groups were highly similar to those of CRC-related gut microbiota [41,42,43,44]. 

At the species level, *B. animalis* had the highest abundance and has been shown to have the potential to inhibit CRC in a large number of clinical and animal experiments, leading to shortened intestinal recovery time and the effective recovery of gut microbiota diversity after the administration of *B. animalis* milk subsp. MH-02 in patients undergoing endoscopic colonic polypectomy surgery [45]. The abundance of *L. plantarum* increased significantly in the Zhang-LL live bacteria treatment group, indicating that Zhang-LL has some intestinal colonization ability. Recent studies have shown that using the *Lactobacillus* strain, either alone or in combination with standard chemotherapy drugs for CRC, has adverse effects on tumor development from various perspectives, including the induction of apoptosis, immunomodulation, changes in the gut microbiota, and signaling pathways involved in cell migration and invasion, thereby inhibiting tumor metastasis [46]. *L. paracasei* also showed significant changes, but most studies have been conducted at the cellular level. For example, *L. paracasei* K5 exhibited adhesion to Caco-2 colon cancer cells, inducing apoptosis via the expression of the Bcl-2 family of proteins [47]. Heat-killed strains cannot colonize the intestine and do not increase *Lactobacillus* abundance; it is speculated that heat-killed Zhang-LL works by increasing the abundance of other beneficial commensal bacteria. Furthermore, the mouse gut metagenome *Enterorhabdus* showed consistent results with the genus-level dominant flora, but the sequencing results did not provide species-level information for this OTU.

Numerous studies have identified specific probiotic strains with anti-tumor activity, and all have found changes in intestinal structure after strain intervention; however, the effects on specific metabolites have not been defined. Thus, this study aimed to identify metabolite changes associated with CRC progression. Our results showed that following Zhang-LL administration, metabolites were enriched in the cyclooxygenase enzymatic pathway of arachidonic acid metabolism (Appendix A). Many studies have confirmed that arachidonic acid metabolism is associated with the occurrence of various diseases, including cardiovascular disease, myocardial hypertrophy, rheumatoid arthritis, and nonalcoholic fatty liver disease [48,49,50,51]. 

Furthermore, arachidonic acid metabolites play an important role in immune system function, allergy promotion and regression, inflammation, mood, and appetite [52]. The FADS1 enzyme has been shown to induce the enrichment of Gram-negative bacteria through a high arachidonic acid microenvironment, activate the conversion of the TLR4/MYD88 pathway to PGE2, and ultimately promote CRC tumorigenesis [53]. COX-2 and PGE2, which are key factors in the enzymatic pathway of cyclooxygenase, are frequently used as chemoprevention methods for CRC with the COX-2 inhibitor celecoxib, which inhibits adenoma progression in a dose-dependent manner and prolongs survival in vivo [54]. PGE2 induces cancer stem cell expansion via EP4-PI3K and EP4-mitogen-activated protein kinase signaling by activating NF-κB, and it promotes the formation of CRC liver metastases in mice. Similarly, our validation results revealed that PGE2 levels in the intestinal contents were significantly elevated following AOM/DSS induction and significantly reduced following Zhang-LL intervention, indicating that the breakdown process of arachidonic acid metabolism was inhibited. Although there was no consistent change in the different PG contents and PGE2 detected in the sequencing results, the sensitivity of the detection method, the detection range, and the different test sample choices must all be considered.

Finally, we investigated the effect of Zhang-LL intervention on the expression of genes involved in immune system regulation. According to the transcriptomics Reactome functional enrichment analysis results, both the Zhang-LL viable bacteria and heat-inactivation treatment groups were enriched in genes related to the CD22-mediated BCR regulatory pathway. By negatively regulating BCR signaling, CD22, an inhibitory BCR co-receptor, sets the BCR signaling threshold and prevents B-cell overstimulation [55]. B cells perceive the microenvironment via B-cell receptors (BCR) and toll-like receptors (TLRs); CD22 negatively regulates TLR signals, which have a receptor modulation role in mediating B-cell adaptive and innate immune responses [56]. CD22-mediated pathways have not been enriched in CRC prevention and treatment; however, CD22 has been used as an immunotherapy target in the treatment of tumors such as triple-negative breast cancer, liver cancer, and B-cell lymphoblastic lymphoma [57,58,59]. 

It was also found that CD22 ligand destruction in CD45 knockout B cells increased CD22 phosphorylation, but phosphorylation did not occur in B-cell mutants expressing SHP-1 loss of function, indicating that SHP-1 rather than CD45 is required for ligand-mediated CD22 regulation, providing further evidence that CD22 is the substrate of SHP-1 [60]. Regorafinib was developed as an immunotherapeutic drug for CRC as a SHP-1 target [61]. Another study showed that RNF6 increased STAT3 phosphorylation by promoting SHP-1 ubiquitination and degradation, demonstrating that the RNF6/SHP-1/STAT3 axis might be a potential treatment option for tumors. However, studies of the mechanism of action for mitigating CRC progression have been relatively concentrated in the signaling pathways associated with STAT3 and the recruitment of mediated immune cells [62,63]. Transcriptomic KEGG functional enrichment analysis showed that signaling pathways directly related to CRC progression, such as the Toll-like receptor, NOD-like receptor, mTOR, and Hippo signaling pathways, were significantly enriched after Zhang-LL intervention. These findings are consistent with the conclusions of the majority of previous studies, and the relevant signaling pathways have been confirmed to regulate CRC progression through individual verification experiments [64,65,66,67].

The multi-omics joint analysis showed that Zhang-LL intervention promoted the decomposition of arachidonic acid by regulating the abundance of the dominant flora *A. muris* and *L. johnsonii* in the gut microbiota and upregulating the expression of relevant genes in the CD22-mediated BCR regulation process. The strongest positive correlation with arachidonic acid metabolism was found in *Akkermansia*. As an emerging next-generation probiotic, *Akkermansia* is closely associated with the occurrence of gastrointestinal diseases; however, its safety remains unknown [68]. In CRC studies, the correlation between *Akkermansia* and arachidonic acid metabolism was not supported by data, and the current study included mainly live or pasteurized bacteria *A. muciniphila*, which has been shown to effectively reduce hyperuricemia by regulating uric acid metabolism and inflammation [69]. It has also been shown that *A. muciniphila* in combination with quercetin improves early obesity and non-alcoholic fatty liver disease by regulating gut microbiota and bile acid metabolism [70], *A. muciniphila* in the intestine regulates the metabolism of L-aspartate through the intestine-liver axis and improves metabolic dysfunction-associated fatty liver disease and other related metabolic pathways [71]. There is a strong correlation between *A. muciniphila* and CRC progression, and numerous studies have confirmed the anti-tumor effects of *A. muciniphila*. For example, the *A. muciniphila* aspartate protease Amuc_1434* inhibits LS174T colon cancer cell viability via the TRAIL-mediated apoptotic pathway [72]. In addition, purified membrane proteins from *A. muciniphila* attenuated colitis-associated tumorigenesis by modulating CD8 + T cells in mice.

In general, *A. muciniphila* primarily delays the progression of diabetes, obesity, nonalcoholic steatohepatitis, inflammatory bowel disease, multiple sclerosis, cancer, and other diseases by improving the clinical response to checkpoint inhibitor immunotherapy, maintaining intestinal barrier function, and improving the inflammatory response, opening up new paths for precision medicine [73]. However, a few adverse outcomes have shown that the *A. muciniphila*-mediated reconstitution of gut microbiota after antibiotic administration negatively affects colitis-associated CRC progression in mice [74]. On the other hand, *A. muris* research is extremely limited, with information indicating that *Acutalibacter* belongs to Ruminococcaceae, which is found in the gastrointestinal tracts of various mammals, and that *A. muris* is the only species of *Acutalibacter* [75]. In addition, only a few studies on cancer prevention and treatment have addressed the anti-tumor effects of *L. johnsonii*. High-resolution microbiome analysis from only one clinical study showed that *Lactobacillus gasseri*, *L. johnsonii*, *Lactobacillus vaginalis*, and *Fusobacterium nucleatum* were associated with oral and oropharyngeal cancer in the saliva of patients who were HPV-positive and -negative and undergoing surgery and chemoradiotherapy [76].

The correlation analysis between the CD22-mediated BCR regulation process and arachidonic acid metabolism provides important ideas for future research. The intestinal environment after intervention by specific strains induces various immune states and involves the related metabolic processes of gut microbiota derivatives, which can be used to further explore the molecular regulatory processes of different inhibitory co-receptors, such as CD22 and lipid metabolism, as well as arachidonic acid metabolism to reveal the precise mechanism of Zhang-LL in reversing CRC outcomes.

There were some limitations in our study. Due to the small number of mice intestinal samples available, intestinal content samples were used for diversity analysis; however, fecal microbiota does not necessarily reflect accurate changes in intestinal flora, and differences in tissue and fecal extracted DNA content must be considered [77]. In addition, the specific biomarkers responsible for the anti-CRC effects of Zhang-LL were not identified, making it difficult to establish CRC biomarkers for hosts with different physiological states. In the future, specific CRC drivers could be designed to investigate specific pathways based on the specificity and sensitivity of biomarkers. Furthermore, our current results are not sufficient to generalize the role of probiotics in the prevention and adjuvant treatment of CRC, and the beneficial effects of *L. plantarum* Zhang-LL cannot be extrapolated to another strain from the same species; we speculate that different strains exert anti-tumor effects through different mechanisms. Before clinical trials can be conducted, additional in vitro and in vivo experiments are required to identify potential probiotic strains and reasonable implementations. Given the higher degree of interaction between multiple derived metabolites and multiple gut microbiota, we also need to investigate probiotics with high specificity in metabolic pathways or gene regulation associated with tumors. The direct effects of locally present microbiota in the tumor microenvironment or their manipulation of distal flora to affect the overall flora structure require further study.

## 5. Conclusions

In conclusion, this study showed that both viable *L. plantarum* Zhang-LL bacteria and heat-killed strains could alleviate the progression of CRC in AOM/DSS mice, and that its mechanism was to inhibit the arachidonic acid metabolism pathway by regulating the abundance of the dominant flora *Lactobacillus* in the gut microbiota and upregulating the expression of genes related to the BCR regulatory process mediated by CD22 to improve host immunogenicity.

## Figures and Tables

**Figure 1 nutrients-15-04512-f001:**
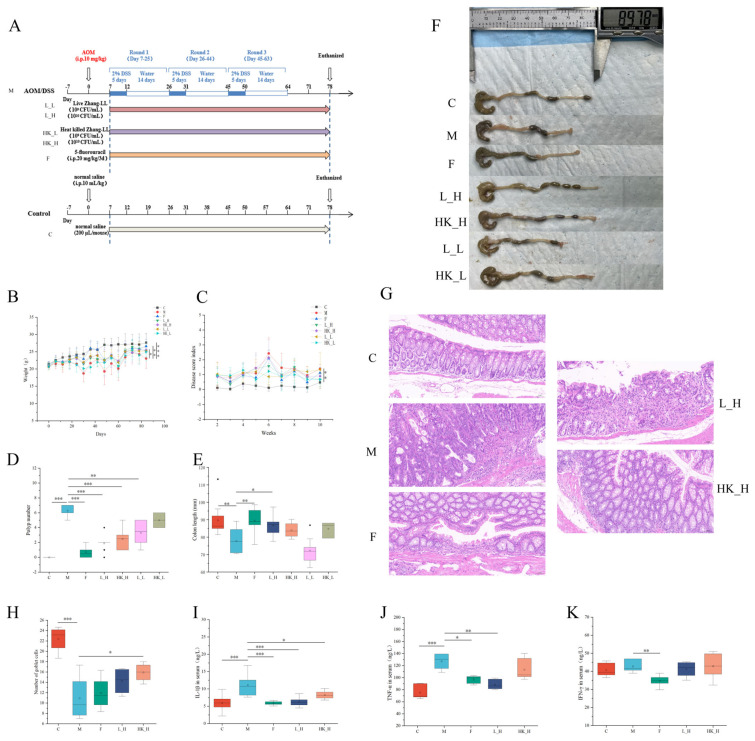
Ameliorative effect of *L. plantarum* Zhang-LL in AOM/DSS-induced colorectal cancer (CRC) mice. F: 5-fluorouracil; L_H: live bacteria in high doses; HK_H: high-dose heat-killed bacteria; L_L: live bacteria in low doses; HK_L: low-dose heat-killed bacteria. (**A**) Schematic diagram of the CRC mice model establishment and intervention methods (L_H, HK_H, L_L, and HK_L). (**B**) The average body weight of mice in different treatment groups (g). (**C**) Disease activity index (DAI) of mice. (**D**) Polyp number. (**E**) Colon length (mm). (**F**) Representative photographs of colons from mice. (**G**) Pathological changes by HE staining (200× magnification) in mice colon tissue. (**H**) The number of mice colon goblet cells in random fields of view. Levels of (**I**) IL-1β, (**J**) TNF-α, and (**K**) IFN-γ in the serum (ng/L). Data are presented as mean ± standard deviation (SD). N = 8. *p* < 0.05, *; *p* < 0.01, **; *p* < 0.001, ***.

**Figure 2 nutrients-15-04512-f002:**
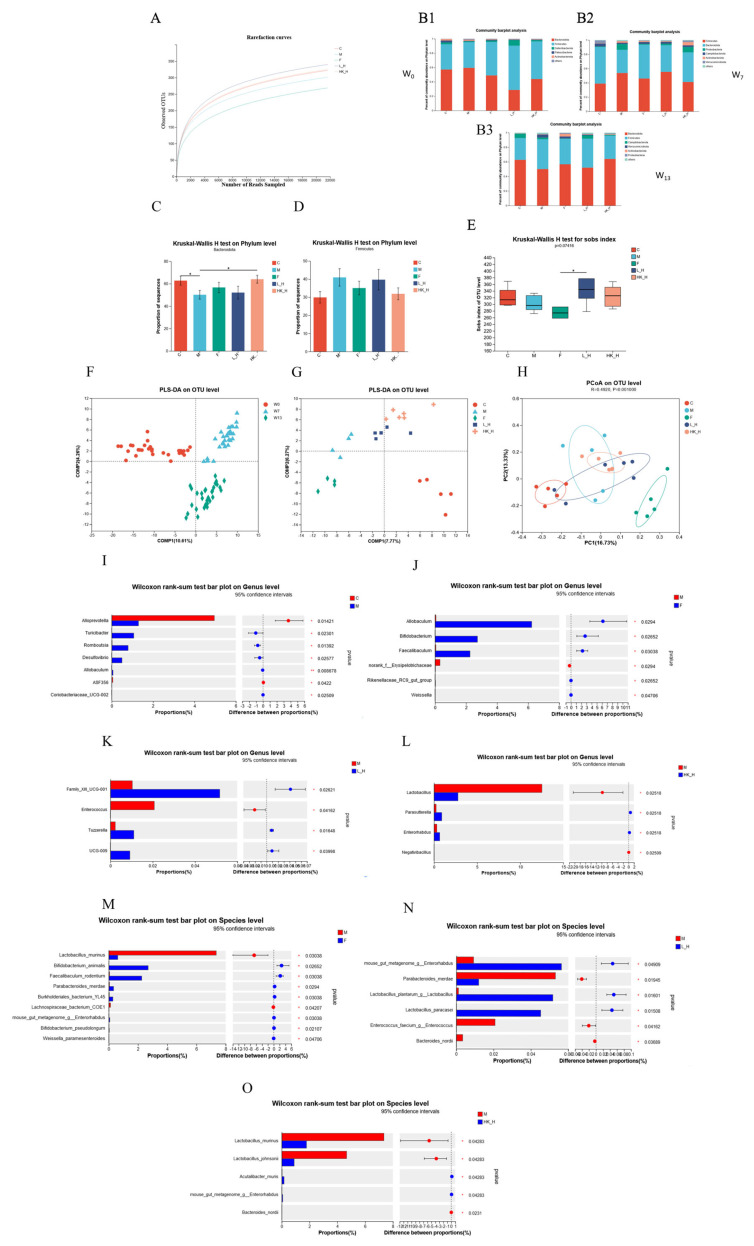
The effects of F, L_H, and HK_H on the composition of gut microbiota. (**A**) Rarefaction curves. Relative abundance of gut microbiota at (**B1**) W_0_, (**B2**) W_7_, and (**B3**) W_13_ for various phyla. Relative abundances of (**C**) *Bacteroidetes* and (**D**) *Firmicutes* between different treatment groups. (**E**) sobs index. (**F**) Partial Least Squares Discriminant Analysis (PLS-DA) intergroup similarity analysis at different time points of modeling and (**G**) after mouse sacrifice. (**H**) Principal coordinates analysis (PCoA) analysis of intergroup similarity after sacrifice. Relative abundance of gut microbiota at the genus level with group (**I**) CNT, (**J**) F, (**K**) L_H, and (**L**) HK_H compared to that of group M. Relative abundance of gut microbiota at species level compared with group (**M**) F, (**N**) L_H, and (**O**) HK_H compared with that of group M. *n* = 6. *p* < 0.05, *.

**Figure 3 nutrients-15-04512-f003:**
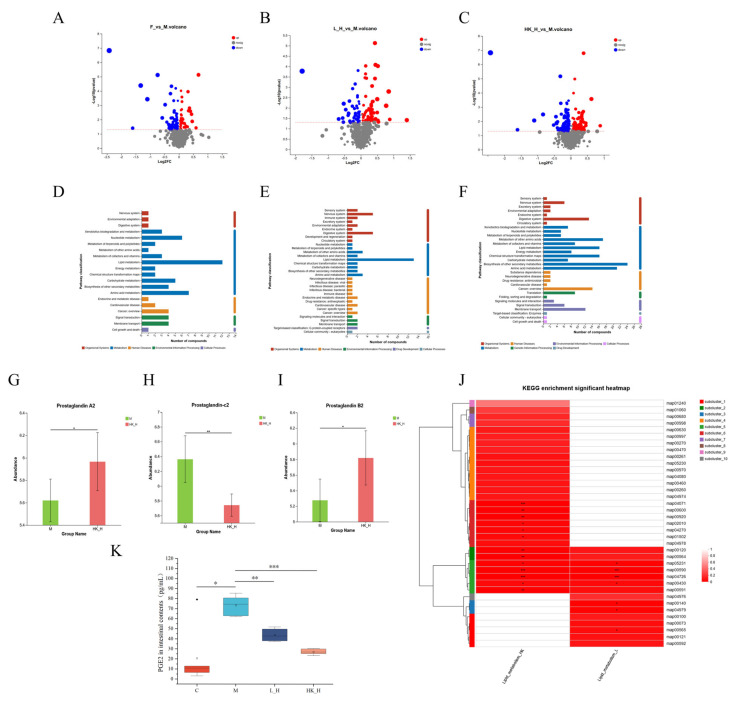
L_H and HK_H alter gut microbiota-related metabolites in AOM/DSS mice. Volcano plot of differential metabolites between mice in the (**A**) group F and group M, (**B**) group L_H and group M, and (**C**) group HK_H and group M. Pathway enrichment analysis of differential metabolites between mice in (**D**) group F and group M, (**E**) group L_H and group M, and (**F**) group HK_H and group M. Differences in the relative abundance of (**G**) PGA2, (**H**) PGC2, and (**I**) PGB2 in groups HK_H and M. (**J**) Lipid metabolism-related pathway enrichment heat map in groups L_H and HK_H. (**K**) Levels of PGE2 (pg/mL) in intestinal contents of mice. Data are presented as mean ± standard deviation (SD). *n* = 6. *p* < 0.05, *; *p* < 0.01, **; *p* < 0.001, ***.

**Figure 4 nutrients-15-04512-f004:**
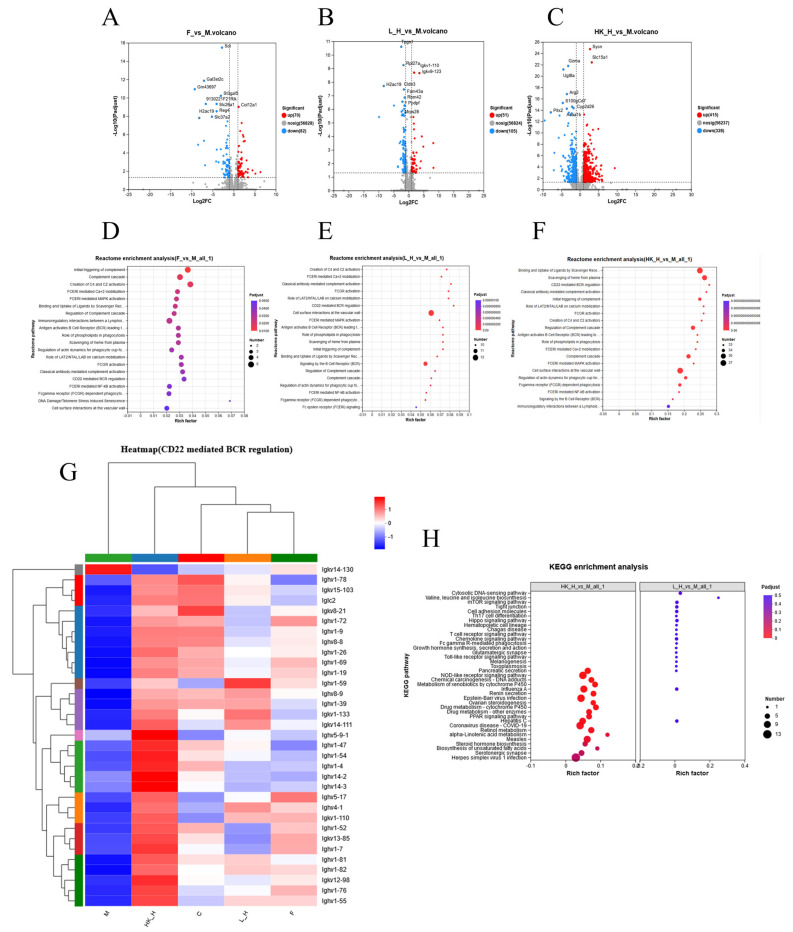
Effects of L_H and HK_H on gene expression in the colon tissues of AOM/DSS mice. Volcano map of differentially expressed genes between mice in the (**A**) group F and group M, (**B**) group L_H and group M, and (**C**) group HK_H and group M. Reactome pathway enrichment analysis of differentially expressed genes between mice in the (**D**) group F and group M, (**E**) group L_H and group M, and (**F**) group HK_H and group M. (**G**) Cluster analysis of genes associated with the CD22-mediated BCR regulation pathway. (**H**) KEGG pathway enrichment analysis of differentially expressed genes between groups L_H, HK_H, and M. *n* = 4.

**Figure 5 nutrients-15-04512-f005:**
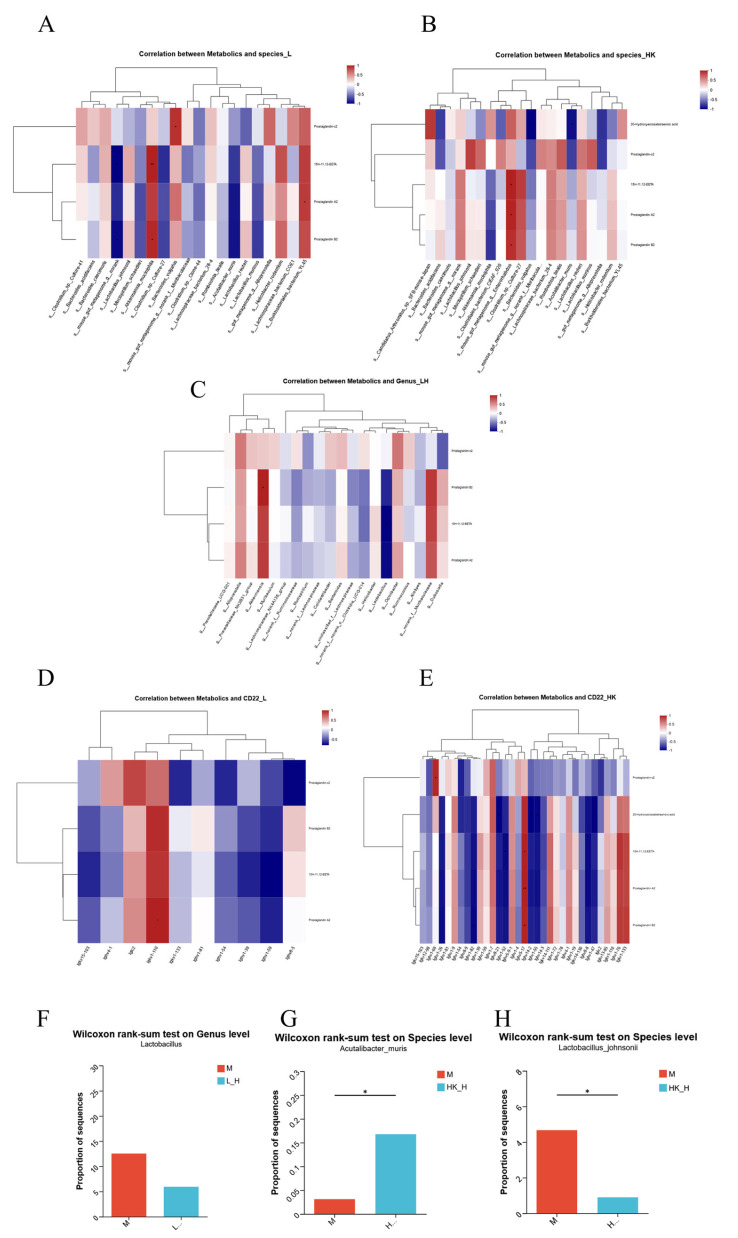
Multi-omics joint analysis of the effects of L_H and HK_H on gut microbiota, metabolites, and gene expression in AOM/DSS mice. Correlation analysis of gut microbiota at the species level with metabolites in the (**A**) groups L_H and (**B**) HK_H. (**C**) Correlation analysis of gut microbiota at the genus level with metabolites in group L_H. Correlation analysis of metabolites with differentially expressed genes in groups (**D**) L_H and (**E**) HK_H. Differences in the relative abundance of (**F**) *Lactobacillus*, (**G**) *A. muris*, and (**H**) *L. johnsonii*. *p* < 0.05, *; *p* < 0.01, **.

## Data Availability

The data presented in this study are available upon request from the authors.

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
