# Peer review of "Lactobacillus plantarum Zhang-LL Inhibits Colitis-Related Tumorigenesis by Regulating Arachidonic Acid Metabolism and CD22-Mediated B-Cell Receptor Regulation"

_nutrients, 2023, doi:10.3390/nu15214512_

Round 1
Reviewer 1 Report
Zhu et al studied the effect of probiotic L. plantarum Zhang-LL and heat-inactivated strains in colorectal cancer mice. Use of live and heat killed bacteria in treating CRC looks interesting. Dose dependent treatment with both L_H and HK_H and by analyzing changes in the gut microbiota, metabolites and host transcription, the regulatory relationship between specific gut microbiota structures and host gene expression is determined. Authors show that high doses of Zhang-LL bacteria lessen the severity of CRC progression. HK_H treatment prevented goblet cell loss, reduced the number of polyps in mice and regulated the gut microbiota by inhibiting the arachidonic acid catabolic processes. Authors also claim that the Zhang-LL improvises the host immunogenicity by upregulating CD22-mediated B cell receptor regulation-related gene expression. The manuscript is interesting and shows evidence for anti-tumor activity by HK_H dose. However, authors have not provided any evidence for the molecular mechanism by which the Zhang-LL heat inactivated strain shows anti-tumor activity.
Authors have pinpointed the limitations of study in the discussion. For example, biomarkers responsible for anti-CRC activity have not been identified in the current study. Also, the number of animal replicates needs to be increased in order to claim that Zhang-LL bacteria could be used as probiotics.
Apart for the above comments the manuscript is clear, comprehensive and relevant to the cancer treatment field. The flow of the manuscript is smooth. Important articles in the recent past have been included and justified in the manuscript. There are no excessive self-citations. The data provided in the current manuscript is of great importance, relevance and a valuable addition to the existing knowledge of the scientific community.
Specific comments
The data looks promising however the labels are hardly visible. Fig resolution is low and poor. It is difficult to read the X and Y axes labels in Fig 1. For example, in Fig 3, there are too many panels. It could be either reduced to a few and others can go as supplementary files. Fig 4 and 5 also has similar problem. It almost impossible to read the labels for reactome enrichment analysis and correlation between metabolics and genus. Please include pictures with better resolution.
Few of the article references like from number 61-74 show only the first and last letter of the author’s name. Kindly correct this to appropriate reference format.
Author Response
Dear Reviewers,
On behalf of my co-authors, we thank you very much for giving us an opportunity to revise our manuscript, we appreciate editor and reviewers very much for their positive and constructive comments and suggestions on our manuscript .We have studied reviewer’ s comments carefully and have tried our best to revise our manuscript according to the comments. The revised version has been uploaded.

Reviewer 2 Report
The topic of the manuscript is interesting and extensive research has been carried out by authors. However, the study could be described more concisely, and too bad reading the text I couldn't follow figures because of their quality.
The study examines the influence of probiotic bacterial strain, Lactococcus plantarum Zhang-LL on the development of colorectal cancer in mice. The topic is relevant, but I couldn't say that it is novel. There are some studies that already showed the positive effect of lactocooci (and even the same species, L. plantarum) on the treatment of colorectal cancer. However, the study is exceptionally comprehensive, describing the changes of microbiota, metabolites and host immune system after the administration of L. plantarum.The results are promissing and described in very detail, unfortunately the figures are of bad quality and it's impossible to see what they are depicting. Thus, one can rely only on what is written.
Author Response

(The authors gave the same response as above.)

Reviewer 3 Report
I love the topic of this manuscript. The probiotic and postbiotic of L. plantarum Zhang-LL active and heat-killed cells gavage comparison is very important to help this research area understand the function resource of the microbiome. I have a big interest in reading the manuscript and looking their sufficient results from histopathology, gut microbiome communities, untargeted metabolomics, and mice intestinal tissue transcriptomics. However, the small characters on the figures stopped me to do that. I hope the authors can remake the figures and enlarge the characters in the figures to let me keep reading the manuscript and learning about their study.
Also, for the M&M 2.2, if the authors can add the different group names in Fig 1A, it will help a lot for the audiences to understand the study design!
I am looking forward to receiving the revised manuscript as soon as possible.
Author Response
Dear Reviewers,
On behalf of my co-authors, we thank you very much for giving us an opportunity to revise our manuscript, we appreciate editor and reviewers very much for their positive and constructive comments and suggestions on our manuscript. We have studied reviewer’ s comments carefully and have tried our best to revise our manuscript according to the comments. The revised version has been uploaded.

Round 2
Reviewer 3 Report
I cannot read the figures. I tried to enlarge the figures to 300%, but all the figure legends are Vague. I don't know which line is which data. Thus, I cannot make any comments on this manuscript! The authors should go through all the figures and make the figure legends readable!
Author Response
Dear reviewer,
On behalf of my co-authors, we thank you very much for giving us an opportunity to revise our manuscript. The revised version has been uploaded.
Thank you for your consideration. I look forward to hearing from you.
Sincerely,
Yuanhong Xie
Beijing University of Agriculture, Beijing, China
102206
13581907345
xieyuanh@163.com

Round 3
Reviewer 3 Report
Thanks for the author's work to improve the quality of the figures. All the figures are readable now. There are some questions I think the authors need to answer and modify to hit the published standard:
1. With the inoculation of live Zhang-LL, is there any observation of portion change/existence of Lactobacillus plantarum Zhang-LL in the 16S Illumina data? In the data, I see different Lactobacillus, but none is Lactobacillus plantarum. Thus, the authors need to clarify it.
2. For the 16S illumina sequencing, the contents from which part of the intestine were collected? duodenum, jejunum or ileum?
3. In Figure 1A, the mice were euthanized on day 78, how did the authors collect the week 13 microbiome? Please explain.
4. What are the key metabolites of Lactobacillus? Is there any difference in metabolites between live and heat-killed lactobacillus?
5. The link between microbial community and metabolites. I think the authors need to point out the direct effect of L. plantarum Zhang-LL on the shifting of metabolites.
Author Response
Dear Reviewer,
On behalf of my co-authors, we thank you very much for giving us an opportunity to revise our manuscript, we appreciate editor and reviewers very much for their positive and constructive comments and suggestions on our manuscript. We have studied reviewer’ s comments carefully and have tried our best to revise our manuscript according to the comments. The revised version has been uploaded.
- With the inoculation of live Zhang-LL, is there any observation of portion change/existence of Lactobacillus plantarum Zhang-LL in the 16S Illumina data? In the data, I see different Lactobacillus, but none is Lactobacillus plantarum. Thus, the authors need to clarify it.
A:As shown in Figure 2N, the comparison of Zhang-LL live bacteria and model groups at the species level showed that Zhang-LL effectively increased the relative abundance of Lactiplantibacillus plantarum and Lactobacillus paracasei after intervention.
- For the 16S illumina sequencing, the contents from which part of the intestine were collected? duodenum, jejunum or ileum?
A:Samples were derived from the contents of the colon and rectum.
- In Figure 1A, the mice were euthanized on day 78, how did the authors collect the week 13 microbiome? Please explain.
A:The text is incorrectly stated here, and the 78th day and the 13th week are the same point in time. The formal experiment totaled 78 days, while 13 weeks additionally included a one-week adaptation period for the animals.
- What are the key metabolites of Lactobacillus? Is there any difference in metabolites between live and heat-killed lactobacillus?
A:Key metabolites of lactobacillus include Bacteriocins, Enzymes, Bacterial cell wall, Short-chain fatty acids, and others. Heat-killed strains do not represent a difference between their metabolites and those of viable bacteria. If there is a difference in the mechanism, it is mainly due to the colonization of live bacteria, which affects host metabolism.
- The link between microbial community and metabolites. I think the authors need to point out the direct effect of L. plantarum Zhang-LL on the shifting of metabolites.
A:It has been shown in a large number of studies that beneficial intestinal commensal bacteria are closely related to metabolites such as amino acids, short-chain fatty acids, bile acids, sphingolipids and hydrogen sulfide, and this study proves that Zhang-LL intervenes in regulating arachidonic acid metabolism and identifying Lactobacillus with a high positive correlation. Therefore, it can be shown that Zhang-LL can directly affect arachidonic acid metabolism, especially downstream prostaglandin metabolism pathways.